# Convergence Analysis of an Inexact Three-Operator Splitting Algorithm

**Chunxiang Zong [1], Yuchao Tang [1,\*] and Yeol Je Cho [2,3]**

[1] Department of Mathematics, NanChang University, Nanchang 330031, China; zongchunxiang@email.ncu.edu.cn

[2] School of Mathematical Sciences, University of Electronic Science and Technology of China, Chengdu 611731, China; yjchomath@gmail.com

[3] Department of Mathematics Education, Gyeongsang National University, Jinju 52828, Korea

[\*] Correspondence: hhaaoo1331@163.com or yctang@ncu.edu.cn

**Abstract:** The three-operator splitting algorithm is a new splitting algorithm for finding monotone inclusion problems of the sum of three maximally monotone operators, where one is cocoercive. As the resolvent operator is not available in a closed form in the original three-operator splitting algorithm, in this paper, we introduce an inexact three-operator splitting algorithm to solve this type of monotone inclusion problem. The theoretical convergence properties of the proposed iterative algorithm are studied in general Hilbert spaces under mild conditions on the iterative parameters. As a corollary, we obtain general convergence results of the inexact forward-backward splitting algorithm and the inexact Douglas-Rachford splitting algorithm, which extend the existing results in the literature.

**Keywords:** inexact three-operator splitting algorithm; nonexpansive operator; fixed point

---

## 1. Introduction

Operator splitting algorithms have been widely applied for solving various convex optimization problems in signal and image processing, medical image reconstruction, machine learning and others. In particular, the forward-backward splitting algorithm [1], the Douglas-Rachford splitting algorithm [2,3] and Tseng's forward-backward-forward splitting algorithm [4] are classical operator splitting algorithms for solving monotone inclusion problems. Many popular optimization algorithms can be derived from them, such as the proximal gradient algorithm [5,6], the primal-dual fixed point algorithm [7,8], the alternating directions method of multipliers (ADMM) [9–11], the primal-dual splitting algorithm [12–16] and many others (for more results, see [17–19] for a comprehensive review). Recently, to solve large-scale optimization problems, new operator splitting algorithms were proposed, that is, algorithms including the generalized forward-backward splitting algorithm [20,21], the variable metric forward-backward splitting algorithm [22], the asymmetric forward-backward-adjoint splitting algorithm [23] and the inertial forward-backward splitting algorithm [24].

In particular, Davis and Yin [25] proposed a three-operator splitting algorithm to solve the following monotone inclusion problem,

$$\text{Find } x \in H \text{ such that } 0 \in Ax + Bx + Cx, \tag{1}$$

where $H$ is a real Hilbert space, $A : H \to 2^H$ and $B : H \to 2^H$ are maximally monotone operators, and $C : H \to H$ is a cocoercive operator. They pointed out that the three-operator

splitting algorithm includes not only the forward-backward operator splitting algorithm [26] and the Douglas-Rachford operator splitting algorithm [27], but also the forward-Douglas-Rachford operator splitting algorithm [28]. Also, they proved some of the convergence theorems of the three-operator splitting algorithm under mild conditions on the parameters and studied the convergence rates of the iterative algorithm. It is worth mentioning that Raguet et al. [20] introduced a generalized forward-backward splitting (GFBS) algorithm for solving monotone inclusion of the sum of a finite family of maximally monotone operators and a cocoercive operator. It is known that the sum of a finite family of maximally monotone operators can be represented by the sum of two maximally monotone operators in a suitable product space, where one of them is the normal cone of a closed vector subspace. Therefore, the GFBS algorithm could be recovered by the three-operator splitting algorithm. Conversely, it is not clear how to deduce the three-operator splitting algorithm from the GFBS algorithm. Besides, a preconditioning extension of the GFBS algorithm was developed by Raguet and Landrieu [21].

On the other hand, it is worth mentioning that Vũ [15] introduced a primal-dual splitting algorithm to solve the monotone inclusion with the sum of mixtures of maximally monotone operators, parallel sums of maximally monotone operators with linear operators, and cocoercive operators. Although this general monotone inclusion includes the three-operator monotone inclusion (1) studied in Davis and Yin [25], the obtained primal-dual splitting algorithm introduced additional dual variables. Therefore, the primal-dual splitting algorithm is different from the three-operator splitting algorithm.

The corresponding convex optimization problem related with the three-operator inclusion problem (1) is as follows:

$$\min_{x \in H} \ f(x) + g(x) + h(x), \tag{2}$$

where $g, h : H \to (-\infty, +\infty]$ are proper, lower-semicontinuous convex functions and $f : H \to \mathbb{R}$ is a convex continuous differentiable function, and its gradient $\nabla f$ is $L$-Lipschitz continuous, for some $L \in (0, +\infty)$. Under the assumptions that the proximity operators of $g$ and $h$ have an explicit closed-form solution, the three-operator splitting algorithm [25] can be applied directly to solve the convex minimization problem (2) by letting $A = \partial g$, $B = \partial h$ and $C = \nabla f$, where $\partial g$ and $\partial h$ denote the subdifferentials of $g$ and $h$, respectively. The convex optimization problem (2) includes many real problems that have appeared in signal and image processing, material sciences and medical image reconstruction, etc. See, for example [6,29–31].

Since the three-operator splitting algorithm [25] is a new algorithm, there exists relatively few works directly related to it. Cevher et al. [32] extended the three-operator splitting algorithm [25] from deterministic setting to stochastic setting for solving the monotone inclusion problem (1). Further, Yurtsever et al. [33] proposed a stochastic three-composite minimization algorithm (S3CM) for solving the convex minimization problem of the sum of three convex functions (2). Besides, Pedregosa and Gidel [34] proposed a novel adaptive three-operator splitting algorithm, which choses the step-size without knowing the Lipschitz constant of the gradient operator. However, in these works, they did not consider errors that appeared in the computation of proximity operators or gradient operators.

As we have mentioned before, operator splitting algorithms provide a simple way to construct an effective iterative algorithm for solving many structure convex optimization problems, for example (2). It is sufficient to require that the proximity operator of the corresponding function has an explicit closed-form solution. However, there are also many convex functions which exist that do not have a closed-form solution of their proximity operators. Although it is difficult to compute the exact proximity operator for these functions, it can be accurately calculated under certain error conditions. In general, the operator splitting algorithm that allows the calculation of the proximity operator with error is called an inexact operator splitting algorithm. These were developed together with the exact operator splitting algorithms. We will briefly review some existing works for inexact operator splitting schemes. The well-known inexact proximal point algorithm was first studied in Rockafellar [35].

In [27], Eckstein and Bertsekas proposed a relaxed inexact proximal point algorithm and a relaxed inexact Douglas-Rachford splitting algorithm.

In the context of convex minimization problem, Combettes and Wajs [26] introduced an inexact forward-backward splitting algorithm. They analyzed the convergence of the algorithm, which was based on the fixed point theoretic framework in [36]. He and Yuan [37] and Salzo and Villa [38] studied an accelerated inexact proximal point algorithm ($f = 0$ and $g = 0$ in (2)). Subsequently, an accelerated inexact forward-backward splitting algorithm was proposed by Villa et al. [39], who not only proved that the objective function values have a convergence rate $1/k^2$ when the allowable error is a certain type, but also presented a global analysis of iteration-complexity. Schmidt et al. [40] analyzed the convergence rates of an accelerated proximal-gradient algorithm with an inexact proximity operator. Inspired by a special case of the hybrid proximal outer gradient method of Solodov and Svaiter [41], Eckstein and Yao [42] derived an inexact Douglas-Rachford operator splitting algorithm and an inexact ADMM operator algorithm. Recently, the algorithm was further developed by Alves and Geremia [43], who proved complexity of the inexact Douglas-Rachford algorithm (for more results on other inexact operator splitting algorithms, see [43–52]).

The purpose of this paper is to introduce an inexact three-operator splitting algorithm (Algorithm 1) to solve the monotone inclusion (1). The corresponding resolvent operators and cocoercive operator are allowed to be computed with errors. Under mild conditions with the parameters and errors, we investigate the convergence behavior of the inexact three-operator splitting algorithm. Furthermore, we recover the inexact forward-backward splitting algorithm and the inexact Douglas-Rachford splitting algorithm as corollaries.

The rest of this paper is organized as follows. In Section 2 we review some background on monotone operators and convex analysis. In Section 3, we present the inexact three-operator splitting algorithm and its convergence theorem. Finally, we give some conclusions and future works.

---

**Algorithm 1:** An inexact three-operator splitting algorithm

---

**Input:** For arbitrary $z^0 \in H$, choose $\gamma$ and $\lambda_k$.
    For each $k = 0, 1, 2, \cdots$, compute

1: $x_B^k = J_{\gamma B}(z^k) + e_B^k$;
2: $x_A^k = J_{\gamma A}(2x_B^k - z^k - \gamma(Cx_B^k + e_C^k)) + e_A^k$;
3: $z^{k+1} = z^k + \lambda_k(x_A^k - x_B^k)$.
    Stop when a given stopping criterion is met.

**Output:** $x_B^k, x_A^k$ and $z^{k+1}$.

---

## 2. Preliminaries

In this paper, let $H$ be a real Hilbert space. The inner product and the associated norms of $H$ are denoted by $\langle,\rangle$ and $\|\cdot\|$, respectively. Let $\Gamma_0(H)$ denotes the class of proper, lower semicontinuous and convex functions from $H$ to $(-\infty, +\infty]$. Let $Fix(T)$ denotes the fixed points set of an operator $T$. We use the symbols $\rightharpoonup$ and $\rightarrow$ to denote weak and strong convergence, respectively.

The following definitions and properties are mostly found in [53].

**Definition 1.** *(Zeros, Domain, Range, Graph and Resolvent) Let $A : H \rightarrow 2^H$ be a set-valued operator, where $2^H$ denotes the power set of H. Let I be the identity operator on H. Then,*

(1) *The set of zeros of A is* zer $A := \{x \in H : 0 \in Ax\}$;
(2) *The domain of A is* dom $A := \{x \in H : Ax \neq \varnothing\}$;
(3) *The range of A is* ran $A := \{y \in H : \exists x \in H : y \in Ax\}$;
(4) *The graph of A is* gra $A := \{(x, y) \in H \times H : y \in Ax\}$;
(5) *The resolvent of A is* $J_A = (I + A)^{-1}$.

**Definition 2.** *(Maximal monotone operator) Let $A : H \to 2^H$ be a set-valued operator. Then A is said to be monotone if*

$$\langle x - y, u - v \rangle \geq 0, \quad \forall (x, u) \in gra\, A, (y, v) \in gra\, A. \tag{3}$$

*A is said to be maximally monotone if there exists no monotone operator $B : H \to 2^H$ such that the graph of B properly contains gra A.*

**Definition 3.** *(Nonexpansive and α-averaged) Let $T : H \to H$ be an operator, T is said to be nonexpansive if*

$$\|Tx - Ty\| \leq \|x - y\|, \quad \forall x, y \in H. \tag{4}$$

*Let $\alpha \in (0, 1)$, T is said to be α-averaged if there exists a nonexpansive operator R such that $T = (1 - \alpha)Id + \alpha R$. If $\alpha = \frac{1}{2}$, then T is called a firmly nonexpansive operator.*

The following two lemmas give some useful characterizations of firmly nonexpansive operators and α-averaged operators.

**Lemma 1.** *Let $T : H \to H$ be an operator. Then, the following statements are equivalent:*

(1) *T is firmly nonexpansive.*
(2) *$2T - Id$ is nonexpansive.*
(3) *For all $x, y \in H$, $\|Tx - Ty\|^2 \leq \langle Tx - Ty, x - y \rangle$.*

**Lemma 2.** *Let $T : H \to H$ is nonexpansive, and let $\alpha \in (0, 1)$. Then the following are equivalent:*

(1) *T is α-averaged.*
(2) *$(1 - \frac{1}{\alpha})Id + (\frac{1}{\alpha})T$ is nonexpansive.*
(3) *For all $x, y \in H$, $\|Tx - Ty\|^2 \leq \|x - y\|^2 - \frac{1-\alpha}{\alpha}\|(Id - T)x - (Id - T)y\|^2$.*

**Lemma 3.** *Let $\gamma \in (0, +\infty)$. Let $A : H \to 2^H$ be a maximally monotone operator. Then $J_{\gamma A} : H \to H$ and $Id - J_{\gamma A} : H \to H$ are firmly nonexpansive and maximally monotone.*

**Definition 4.** *(Cocoercive operator) An operator $B : H \to H$ is said to be β-cocoercive with $\beta \in (0, +\infty)$ if*

$$\beta \|Bx - By\|^2 \leq \langle Bx - By, x - y \rangle, \quad \forall x, y \in H. \tag{5}$$

*A cocoercive operator is also called an inverse strongly monotone operator (see, for example, [54]).*

It is easy to see that a β-cocoercive operator is $1/\beta$-Lipschitz continuous.
Let's recall the definition of a uniformly monotone operator.

**Definition 5.** *(Uniformly monotone operator) A set-valued operator $A : H \to 2^H$ is said to be uniformly monotone of a modulus $\phi : [0, +\infty] \to [0, +\infty]$ if $\phi$ is a nondecreasing function with $\phi(0) = 0$ such that*

$$\langle u - v, x - y \rangle \geq \phi(\|x - y\|), \quad \forall (x, u) \in gra\, A, (y, v) \in gra\, A. \tag{6}$$

*If $\phi \equiv \beta(\cdot)^2 > 0$, then A is said to be strongly monotone.*

**Definition 6.** *(Demiregular) A set-valued operator $A : H \to 2^H$ is said to be demiregular at $x \in dom\, A$, if for all $u \in Ax$ and for all sequences $(x^k, u^k) \in gra\, A$ with $x^k \rightharpoonup x$ and $u^k \to u$, we have $x^k \to x$.*

We shall make full use of the following lemma to prove the weak convergence of the iterative sequence.

**Lemma 4.** *Let sequence $\{x_n\} \subset H$. Then we say that $\{x_n\}$ converges weakly to a point $x \in H$ if and only if it is bounded and the weak sequential cluster point is unique.*

The following lemma will be needed in the next section.

**Lemma 5.** *Let $x, y \in H$ and $\lambda \in R$. Then we have*

$$\|\lambda x + (1 - \lambda)y\|^2 + \lambda(1 - \lambda)\|x - y\|^2 = \lambda\|x\|^2 + (1 - \lambda)\|y\|^2. \tag{7}$$

We recall the following lemma, which is crucial to our convergence analysis (See Lemma 5.1 of Combettes [36]):

**Lemma 6.** (Inexact Krasnosel'skiĭ-Mann algorithm) *Let $T : H \rightarrow H$ be nonexpansive, $\{\lambda_n\}$ be a sequence in $(0, 1)$ and $\{e_n\}$ be a sequence in $H$. Suppose that $Fix(T) \neq \emptyset$, $\sum_{n=0}^{+\infty} \lambda_n(1 - \lambda_n) = +\infty$ and $\sum_{n=0}^{+\infty} \lambda_n\|e_n\| < +\infty$. Let $x_0 \in H$, and set*

$$x_{n+1} = x_n + \lambda_n(Tx_n + e_n - x_n), \quad \forall n \geq 0, \tag{8}$$

*Then the following hold:*

*(1)* $\{x_n\}$ *is Fejér monotone with respect to $Fix(T)$.*
*(2)* $\{Tx_n - x_n\}$ *converges strongly to 0.*
*(3)* $\{x_n\}$ *converges weakly to a point in $Fix(T)$.*

When $e_n \equiv 0$, the inexact Krasnosel'skiĭ-Mann algorithm **(iKM)** (8) reduces to the classical Krasnosel'skiĭ-Mann algorithm **(KM)**. The algorithms **(KM)** and **(iKM)** are useful for finding fixed points of nonexpansive mappings. They provide a unified way for analyzing the convergence of various operator splitting algorithms and convex optimization algorithms.

Lemma 6 could be easily extended to the setting of $\alpha$-averaged operators.

**Lemma 7.** *Let $T : H \rightarrow H$ be $\alpha$-averaged, $\{\lambda_n\}$ be a sequence in $(0, \frac{1}{\alpha})$ and $\{e_n\}$ be a sequence in $H$. Suppose that $Fix(T) \neq \emptyset$, $\sum_{n=0}^{+\infty} \lambda_n(\frac{1}{\alpha} - \lambda_n) = +\infty$ and $\sum_{n=0}^{+\infty} \lambda_n\|e_n\| < +\infty$. Let $x_0 \in H$ and set*

$$x_{n+1} = x_n + \lambda_n(Tx_n + e_n - x_n), \quad \forall n \geq 0, \tag{9}$$

*Then the following hold:*

*(1)* $\{x_n\}$ *is Fejér monotone with respect to $Fix(T)$.*
*(2)* $\{Tx_n - x_n\}$ *converges strongly to 0.*
*(3)* $\{x_n\}$ *converges weakly to a point in $Fix(T)$.*

**Proof.** Set $T = (1 - \alpha)I + \alpha S$, where $S$ is nonexpansive. Then the iterative algorithm (9) can be rewritten as

$$x_{n+1} = x_n + \alpha\lambda_n(Sx_n + \frac{1}{\alpha}e_n - x_n) \tag{10}$$

for all $n \geq 0$. Since $Fix(T) = Fix(S)$, it is easy to check that all the conditions of Lemma 6 are satisfied. Thus the results of Lemma 7 follow directly from Lemma 6. This completes the proof. □

## 3. An Inexact Three-Operator Splitting Algorithm

In this section, first, we present an inexact three-operator splitting algorithm. Second, we prove the convergence of it.

Now, we are ready to prove the main convergence results of Algorithm 1. Theorem 1 is parallel to the convergence results of the three-operator splitting algorithm [25], in which the error terms $e_A^k$, $e_B^k$ and $e_C^k$ are equal to zeros in Algorithm 1.

**Theorem 1.** *Let $A : H \to 2^H$ and $B : H \to 2^H$ be maximally monotone operators. Let $C : H \to H$ be a $\beta$-cocoercive operator, for some $\beta > 0$. Assume that $\mathrm{zer}(A + B + C)$ is nonempty. Let $\gamma > 0$ and define an operator $T : H \to H$ as follows,*

$$T := I - J_{\gamma B} + J_{\gamma A}(2J_{\gamma B} - I - \gamma C J_{\gamma B}). \tag{11}$$

*Let $\alpha = \frac{1}{2-\varepsilon}$, where $\varepsilon \in (0,1)$. Assume that $\gamma \in (0, 2\beta\varepsilon)$ and $\lambda_k \in (0, \frac{1}{\alpha})$ such that $\sum_{k=0}^{+\infty} \lambda_k(\frac{1}{\alpha} - \lambda_k) = +\infty$. Let $\{z^k\}$, $\{x_B^k\}$ and $\{x_A^k\}$ be the iterative sequences generated by Algorithm 1. Assume that*

$$\sum_{k=0}^{+\infty} \|e_A^k\| < +\infty, \quad \sum_{k=0}^{+\infty} \|e_B^k\| < +\infty, \quad \sum_{k=0}^{+\infty} \|e_C^k\| < +\infty.$$

*Then the following hold:*

(1)  *$\{z^k\}$ is Fejér-monotone with respect to $\mathrm{Fix}(T)$.*
(2)  *$\{Tz^k - z^k\}$ converges strongly to zero.*
(3)  *$\{z^k\}$ converges weakly to a fixed point of $T$.*
(4)  *If $x^* \in \mathrm{zer}(A + B + C)$, then there exists a constant $M > 0$ such that, for any $\lambda_k \in (0, \frac{1}{\alpha})$,*

$$\sum_{k=0}^{+\infty} \lambda_k \|C J_{\gamma B} z^k - C x^*\|^2 \leq \frac{1}{\gamma(2\beta - \frac{\gamma}{\varepsilon})} \left( \|z^\circ - z^*\|^2 + \sum_{k=0}^{+\infty} \frac{M}{\alpha} \|e^k\| \right). \tag{12}$$

*In addition, we have*

$$\sum_{k=0}^{+\infty} \lambda_k \|C x_B^k - C x^*\|^2 \leq \frac{1}{\gamma(2\beta - \frac{\gamma}{\varepsilon})} \|z^\circ - z^*\|^2 + S, \tag{13}$$

*where*

$$S = \frac{M}{\alpha\gamma(2\beta - \frac{\gamma}{\varepsilon})} \sum_{k=0}^{+\infty} \|e^k\| + \frac{1}{\alpha\beta^2} \sum_{k=0}^{+\infty} \|e_B^k\|(\|e_B^k\| + 2\|z^\circ - z^*\|).$$

(5)  *If $\lambda_k \geq \underline{\lambda} > 0$, then there exists $z^* \in \mathrm{Fix}(T)$ such that the iterative sequence $\{x_B^k\}$ converges weakly to $J_{\gamma B} z^* \in \mathrm{zer}(A + B + C)$.*
(6)  *If $\lambda_k \geq \underline{\lambda} > 0$, then there exists $z^* \in \mathrm{Fix}(T)$ such that the iterative sequence $\{x_A^k\}$ converges weakly to $J_{\gamma B} z^* \in \mathrm{zer}(A + B + C)$.*
(7)  *Let $\lambda_k \geq \underline{\lambda} > 0$ and assume that there exists $z^* \in \mathrm{Fix}(T)$. Suppose that one of the following conditions hold:*

(a)  *$A$ is uniformly monotone on every nonempty bounded subset of $\mathrm{dom}\, A$;*
(b)  *$B$ is uniformly monotone on every nonempty bounded subset of $\mathrm{dom}\, B$;*
(c)  *$C$ is demiregular at every point $x \in \mathrm{zer}(A + B + C)$.*

*Then $\{x_A^k\}$ and $\{x_B^k\}$ converge strongly to $J_{\gamma B} z^* \in \mathrm{zer}(A + B + C)$.*

**Proof.** First, the iterative sequences $\{z^{k+1}\}$ of Algorithm 1 can be written equivalently as

$$
\begin{aligned}
z^{k+1} &= z^k + \lambda_k(x_A^k - x_B^k) \\
&= z^k + \lambda_k(J_{\gamma A}(2x_B^k - z^k - \gamma(Cx_B^k + e_C^k)) + e_A^k - J_{\gamma B}z^k - e_B^k) \\
&= z^k + \lambda_k(Tz^k - z^k + J_{\gamma A}(2x_B^k - z^k - \gamma(Cx_B^k + e_C^k)) \\
&\quad - J_{\gamma A}(2J_{\gamma B}z^k - z^k - \gamma CJ_{\gamma B}z^k) + e_A^k - e_B^k) \\
&= z^k + \lambda_k(Tz^k - z^k + e^k),
\end{aligned}
\tag{14}
$$

where

$$
e^k = J_{\gamma A}(2x_B^k - z^k - \gamma(Cx_B^k + e_C^k)) - J_{\gamma A}(2J_{\gamma B}z^k - z^k - \gamma CJ_{\gamma B}z^k) + e_A^k - e_B^k.
$$

Since $C$ is $\frac{1}{\beta}$-Lipschitz continuous and $J_{\gamma A}$ is nonexpansive, it follows that

$$
\begin{aligned}
\|e^k\| &= \|J_{\gamma A}(2x_B^k - z^k - \gamma(Cx_B^k + e_C^k)) - J_{\gamma A}(2J_{\gamma B}z^k - z^k - \gamma CJ_{\gamma B}z^k) + e_A^k - e_B^k\| \\
&\leq \|J_{\gamma A}(2x_B^k - z^k - \gamma(Cx_B^k + e_C^k)) - J_{\gamma A}(2J_{\gamma B}z^k - z^k - \gamma CJ_{\gamma B}z^k)\| \\
&\quad + \|e_A^k\| + \|e_B^k\| \\
&\leq \|2x_B^k - z^k - \gamma(Cx_B^k + e_C^k) - 2J_{\gamma B}z^k + z^k + \gamma CJ_{\gamma B}z^k\| + \|e_A^k\| + \|e_B^k\| \\
&= \|2e_B^k - \gamma(Cx_B^k + e_C^k) + \gamma CJ_{\gamma B}z^k\| + \|e_A^k\| + \|e_B^k\| \\
&\leq \gamma\|C(J_{\gamma B}z^k + e_B^k) - CJ_{\gamma B}z^k\| + 3\|e_B^k\| + \|e_A^k\| + \gamma\|e_C^k\| \\
&\leq (\tfrac{\gamma}{\beta} + 3)\|e_B^k\| + \|e_A^k\| + \gamma\|e_C^k\|.
\end{aligned}
\tag{15}
$$

Since $\sum_{k=0}^{+\infty}\|e_A^k\| < +\infty$, $\sum_{k=0}^{+\infty}\|e_B^k\| < +\infty$, $\sum_{k=0}^{+\infty}\|e_C^k\| < +\infty$ and $\gamma \in (0, 2\beta\varepsilon)$, we have $\sum_{k=0}^{+\infty}\|e^k\| < +\infty$. Thus, with the help of Proposition 2.1 of [25], $T$ is $\alpha$-averaged and we can obtain the conclusions (1), (2) and (3) by Lemma 7.

(4) Let $x^* \in zer(A + B + C)$. Then, by Lemma 2.2 of [25], there exists $z^* \in Fix(T)$ such that $x^* = J_{\gamma B}(z^*)$. By (1) and $\sum_{k=0}^{+\infty}\|e^k\| < +\infty$, we know that $\{\|z^k - z^*\|\}$ and $\{\|e^k\|\}$ are bounded. In view of (14), Lemma 5 and the Cauchy-schwartz inequality, we have

$$
\begin{aligned}
\|z^{k+1} - z^*\|^2 &= \|z^k + \lambda_k(Tz^k - z^k + e^k) - z^*\|^2 \\
&= \|(1 - \lambda_k)(z^k - z^*) + \lambda_k(Tz^k - z^*) + \lambda_k e^k\|^2 \\
&= \|(1 - \lambda_k)(z^k - z^*) + \lambda_k(Tz^k - z^*)\|^2 + \lambda_k^2\|e^k\|^2 \\
&\quad + 2\lambda_k\langle(1 - \lambda_k)(z^k - z^*) + \lambda_k(Tz^k - z^*), e^k\rangle \\
&\leq \|(1 - \lambda_k)(z^k - z^*) + \lambda_k(Tz^k - z^*)\|^2 + 2\lambda_k\|(1 - \lambda_k)(z^k - z^*) \\
&\quad + \lambda_k(Tz^k - z^*)\|\|e^k\| + \lambda_k^2\|e^k\|^2 \\
&\leq \|(1 - \lambda_k)(z^k - z^*) + \lambda_k(Tz^k - z^*)\|^2 + M\lambda_k\|e^k\| \\
&= (1 - \lambda_k)\|z^k - z^*\|^2 + \lambda_k\|Tz^k - z^*\|^2 \\
&\quad - \lambda_k(1 - \lambda_k)\|Tz^k - z^k\|^2 + M\lambda_k\|e^k\|,
\end{aligned}
\tag{16}
$$

where $M = \sup_{k\in\mathbb{N}}(2\|z^k - z^*\| + \lambda_k\|e^k\|)$. On the other hand, it follows from Remark 2.1 of [25] that

$$
\|Tz^k - z^*\|^2 \leq \|z^k - z^*\|^2 - \frac{1-\alpha}{\alpha}\|Tz^k - z^k\|^2 - \gamma\left(2\beta - \frac{\gamma}{\varepsilon}\right)\|CJ_{\gamma B}z^k - CJ_{\gamma B}z^*\|^2.
\tag{17}
$$

Substituting (17) into (16), we get

$$
\begin{aligned}
\|z^{k+1} - z^*\|^2 \leq \|z^k - z^*\|^2 &- \lambda_k\left(1 - \lambda_k + \frac{1-\alpha}{\alpha}\right)\|Tz^k - z^k\|^2 \\
&- \gamma\lambda_k\left(2\beta - \frac{\gamma}{\varepsilon}\right)\|CJ_{\gamma B}z^k - CJ_{\gamma B}z^*\|^2 + M\lambda_k\|e^k\|,
\end{aligned}
\tag{18}
$$

which implies that

$$
\|z^{k+1} - z^*\|^2 \leq \|z^k - z^*\|^2 - \gamma\lambda_k\left(2\beta - \frac{\gamma}{\varepsilon}\right)\|CJ_{\gamma B}z^k - Cx^*\|^2 + \frac{M}{\alpha}\|e^k\|.
\tag{19}
$$

Then we have

$$
\gamma\lambda_k\left(2\beta - \frac{\gamma}{\varepsilon}\right)\|CJ_{\gamma B}z^k - Cx^*\|^2 \leq \|z^k - z^*\|^2 - \|z^{k+1} - z^*\|^2 + \frac{M}{\alpha}\|e^k\|.
\tag{20}
$$

Summing from zero to infinity, we obtain

$$
\sum_{k=0}^{+\infty} \lambda_k\|CJ_{\gamma B}z^k - Cx^*\|^2 \leq \frac{1}{\gamma(2\beta - \frac{\gamma}{\varepsilon})}\left(\|z^\circ - z^*\|^2 + \sum_{k=0}^{+\infty}\frac{M}{\alpha}\|e^k\|\right).
\tag{21}
$$

Further, we have

$$
\begin{aligned}
\|Cx_B^k - Cx^*\|^2 &= \|Cx_B^k - CJ_{\gamma B}z^k + CJ_{\gamma B}z^k - Cx^*\|^2 \\
&= \|Cx_B^k - CJ_{\gamma B}z^k\|^2 + \|CJ_{\gamma B}z^k - Cx^*\|^2 \\
&\quad + 2\langle Cx_B^k - CJ_{\gamma B}z^k, CJ_{\gamma B}z^k - Cx^*\rangle \\
&\leq \|C(J_{\gamma B}z^k + e_B^k) - CJ_{\gamma B}z^k\|^2 + \|CJ_{\gamma B}z^k - Cx^*\|^2 \\
&\quad + 2\|Cx_B^k - CJ_{\gamma B}z^k\|\|CJ_{\gamma B}z^k - Cx^*\| \\
&\leq \frac{1}{\beta^2}\|e_B^k\|^2 + \|CJ_{\gamma B}z^k - Cx^*\|^2 + \frac{2}{\beta}\|e_B^k\|\|CJ_{\gamma B}z^k - Cx^*\| \\
&\leq \|CJ_{\gamma B}z^k - Cx^*\|^2 + \frac{1}{\beta^2}\|e_B^k\|(\|e_B^k\| + 2\|z^0 - z^*\|).
\end{aligned}
\tag{22}
$$

Therefore, summing (22) from zero to infinity, we obtain

$$
\begin{aligned}
&\sum_{k=0}^{+\infty} \lambda_k\|Cx_B^k - Cx^*\|^2 \\
&\leq \sum_{k=0}^{+\infty} \lambda_k\left(\|CJ_{\gamma B}z^k - Cx^*\|^2 + \frac{1}{\beta^2}\|e_B^k\|(\|e_B^k\| + 2\|z^0 - z^*\|)\right) \\
&\leq \frac{1}{\gamma(2\beta - \frac{\gamma}{\varepsilon})}\left(\|z^\circ - z^*\|^2 + \sum_{k=0}^{+\infty}\frac{M}{\alpha}\|e^k\|\right) + \frac{1}{\alpha\beta^2}\sum_{k=0}^{+\infty}\|e_B^k\|(\|e_B^k\| + 2\|z^\circ - z^*\|).
\end{aligned}
\tag{23}
$$

(5) Since the resolvent operator is firmly nonexpansive, it is also nonexpansive. Then we have

$$
\begin{aligned}
\|x_B^k - J_{\gamma B}z^*\| &= \|J_{\gamma B}z^k + e_B^k - J_{\gamma B}z^*\| \\
&\leq \|J_{\gamma B}z^k - J_{\gamma B}z^*\| + \|e_B^k\| \\
&\leq \|z^k - z^*\| + \|e_B^k\|.
\end{aligned}
\tag{24}
$$

Notice that $\{z^k\}$ and $\{\|e_B^k\|\}$ are bounded, $\{x_B^k\}$ is also bounded. Let $\bar{x}$ be a weak sequential cluster point of $\{x_B^k\}$, say $x_B^{k_n} \rightharpoonup \bar{x}$. From Algorithm 1, it follows that

$$x_B^k - e_B^k = J_{\gamma B} z^k, \quad x_A^k - e_A^k = J_{\gamma A}(2x_B^k - z^k - \gamma(Cx_B^k + e_C^k)).$$

Let

$$u_B^k := \frac{1}{\gamma}(z^k - x_B^k + e_B^k) \in B(x_B^k - e_B^k), \tag{25}$$

and

$$u_A^k := \frac{1}{\gamma}(2x_B^k - z^k - \gamma(Cx_B^k + e_C^k) - x_A^k + e_A^k) \in A(x_A^k - e_A^k). \tag{26}$$

Then we have $(x_B^k - e_B^k, u_B^k) \in \operatorname{gra} B$ and $(x_A^k - e_A^k, u_A^k) \in \operatorname{gra} A$. By (4), we have $Cx_B^k \to Cx^*$, where $x^* \in \operatorname{zer}(A + B + C)$. Since $C$ is cocoercive, it is maximal monotone and so its graph is closed in $H^{weak} \times H^{strong}$, which, together with $x_B^{k_n} \rightharpoonup \bar{x}$, yields $C\bar{x} = Cx^*$. Consequently, we have $Cx_B^{k_n} \to C\bar{x}$. By (2), we have $x_A^k - x_B^k = Tz^k - z^k + e^k \to 0$. Then $x_A^{k_n} \rightharpoonup \bar{x}$. Further, by (3), we obtain

$$u_B^{k_n} \rightharpoonup \frac{1}{\gamma}(z^* - \bar{x}), \quad u_A^{k_n} \rightharpoonup \frac{1}{\gamma}(\bar{x} - z^* - \gamma C\bar{x}).$$

From Corollary 25.5 of [53], we obtain

$$\bar{x} \in \operatorname{zer}(A + B + C), \quad \left(\bar{x}, \frac{1}{\gamma}(z^* - \bar{x})\right) \in \operatorname{gra} B, \quad \left(\bar{x}, \frac{1}{\gamma}(\bar{x} - z^* - \gamma C\bar{x})\right) \in \operatorname{gra} A. \tag{27}$$

Hence $\bar{x} = J_{\gamma B} z^*$. Thus $J_{\gamma B} z^*$ is the unique weak sequential cluster point of $\{x_B^k\}$. By Lemma 4, we can conclude that $\{x_B^k\}$ converges weakly to $J_{\gamma B} z^* \in \operatorname{zer}(A + B + C)$.

(6) Notice that $x_A^k - x_B^k \to 0$ as $k \to +\infty$. By (5), it follows that $\{x_A^k\}$ also converges weakly to $J_{\gamma B} z^* \in \operatorname{zer}(A + B + C)$.

(7) Let $x^* = J_{\gamma B}(z^*)$. By Lemma 2.2 of [25], we have $x^* \in \operatorname{zer}(A + B + C)$. Let $u_B^* := \frac{1}{\gamma}(z^* - x^*)$. Then $u_B^* \in Bx^*$. Let $u_A^* := \frac{1}{\gamma}(x^* - z^*) - Cx^*$. It follow from $x^* \in \operatorname{zer}(A + B + C)$ and $u_B^* \in Bx^*$ that $u_A^* \in Ax^*$.

(a) From (5), we have

$$(x_A^k - e_A^k, u_A^k) \in \operatorname{gra} A, \quad (x_B^k - e_B^k, u_B^k) \in \operatorname{gra} B.$$

Since $B + C$ is monotone, we obtain

$$0 \le \langle x_B^k - e_B^k - x^*, u_B^k + C(x_B^k - e_B^k) - (u_B^* + Cx^*) \rangle. \tag{28}$$

Define $S := \{x^*\} \cup \{x_A^k - e_A^k\}$. By (6), $\{x_A^k\}$ is bounded and so $S$ is also bounded. Since $A$ is uniformly monotone, there exists an increasing function $\Phi_A : R_+ \to [0, +\infty)$ with $\Phi_A(0) = 0$ such that

$$\Phi_A(\|x_A^k - e_A^k - x^*\|) \le \langle x_A^k - e_A^k - x^*, u_A^k - u_A^* \rangle. \tag{29}$$

Adding (28) and (29) yields

$$
\begin{aligned}
&\gamma \Phi_A(\|x_A^k - e_A^k - x^*\|) \\
&\leq \gamma \langle x_A^k - e_A^k - x^*, u_A^k - u_A^* \rangle + \gamma \langle x_B^k - e_B^k - x^*, u_B^k + C(x_B^k - e_B^k) - (u_B^* + Cx^*) \rangle \\
&= \gamma \langle x_A^k - e_A^k - (x_B^k - e_B^k), u_A^k - u_A^* \rangle + \gamma \langle x_B^k - e_B^k - x^*, u_A^k - u_A^* \rangle \\
&\quad + \gamma \langle x_B^k - e_B^k - x^*, u_B^k + C(x_B^k - e_B^k) - (u_B^* + Cx^*) \rangle \\
&= \gamma \langle x_A^k - e_A^k - (x_B^k - e_B^k), u_A^k - u_A^* \rangle + \gamma \langle x_B^k - e_B^k - x^*, u_A^k + u_B^k + C(x_B^k - e_B^k) \rangle \\
&= \langle x_A^k - e_A^k - x_B^k + e_B^k, \gamma u_A^k - \gamma u_A^* \rangle + \langle x_B^k - e_B^k - x^*, x_B^k - x_A^k + e_A^k + e_B^k \\
&\quad + \gamma C(x_B^k - e_B^k) - \gamma C x_B^k - \gamma e_C^k \rangle \\
&= \langle x_A^k - e_A^k - x_B^k, \gamma u_A^k - \gamma u_A^* \rangle + \langle e_B^k, \gamma u_A^k - \gamma u_A^* \rangle \\
&\quad + \langle x_B^k - e_B^k - x^*, x_B^k - x_A^k + e_A^k \rangle + \langle x_B^k - e_B^k - x^*, e_B^k \rangle \\
&\quad + \langle x_B^k - e_B^k - x^*, \gamma C(x_B^k - e_B^k) - \gamma C x_B^k - \gamma e_C^k \rangle \\
&= \langle x_B^k - x_A^k + e_A^k, x_B^k - e_B^k - x^* - \gamma u_A^k + \gamma u_A^* \rangle + \langle e_B^k, x_B^k - e_B^k - x^* + \gamma u_A^k - \gamma u_A^* \rangle \\
&\quad + \langle x_B^k - e_B^k - x^*, \gamma C(x_B^k - e_B^k) - \gamma C x_B^k - \gamma e_C^k \rangle \\
&= \langle x_B^k - x_A^k + e_A^k, z^k - z^* + \gamma C x_B^k - \gamma C x^* + x_A^k - x_B^k - e_A^k - e_B^k + \gamma e_C^k \rangle \\
&\quad + \langle e_B^k, 3x_B^k - z^k - \gamma(C x_B^k + e_C^k) - x_A^k + e_A^k - e_B^k - 2x^* + z^* + \gamma C x^* \rangle \\
&\quad + \langle x_B^k - e_B^k - x^*, \gamma C(x_B^k - e_B^k) - \gamma C x_B^k - \gamma e_C^k \rangle.
\end{aligned}
\tag{30}
$$

By (2) and (4), we have $\|x_B^k - x_A^k\| \to 0$ and $\|C x_B^k - C x^*\| \to 0$ as $k \to +\infty$. Then we have

$$
\langle x_B^k - x_A^k + e_A^k, z^k - z^* + \gamma C x_B^k - \gamma C x^* + x_A^k - x_B^k - e_A^k - e_B^k + \gamma e_C^k \rangle \to 0,
\tag{31}
$$

as $k \to +\infty$. By (3), (5) and (6), we know that $\{x_B^k\}$, $\{x_A^k\}$ and $\{z^k\}$ are bounded. Notice that $\sum_{k=0}^{+\infty} \|e_B^k\| < +\infty$ and $\sum_{k=0}^{+\infty} \|e_A^k\| < +\infty$, we have

$$
\langle e_B^k, 3x_B^k - z^k - \gamma(C x_B^k + e_C^k) - x_A^k + e_A^k - e_B^k - 2x^* + z^* + \gamma C x^* \rangle \to 0,
\tag{32}
$$

as $k \to +\infty$. Since $C$ is $\frac{1}{\beta}$-Lipschitz continuous, we have

$$
\begin{aligned}
\|\gamma C(x_B^k - e_B^k) - \gamma C x_B^k - \gamma e_C^k\| &\leq \gamma \|C(x_B^k - e_B^k) - C x_B^k\| + \gamma \|e_C^k\| \\
&\leq \gamma \frac{1}{\beta} \|e_B^k\| + \gamma \|e_C^k\| \to 0,
\end{aligned}
\tag{33}
$$

as $k \to +\infty$. Then we have

$$
\langle x_B^k - e_B^k - x^*, \gamma C(x_B^k - e_B^k) - \gamma C x_B^k - \gamma e_C^k \rangle \to 0,
\tag{34}
$$

as $k \to +\infty$. From (31), (32) and (34), we obtain

$$
\|x_A^k - e_A^k - x^*\| \to 0,
\tag{35}
$$

as $k \to +\infty$.

Now, we prove that $x_A^k \to x^*$ as $k \to +\infty$. In fact, by (35), we have

$$
\|x_A^k - x^*\| \leq \|x_A^k - x^* - e_A^k\| + \|e_A^k\| \to 0,
\tag{36}
$$

as $k \to +\infty$. Since $x_A^k - x_B^k \to 0$ as $k \to +\infty$. Then we have $x_B^k \to x^*$ as $k \to +\infty$.

(b) Since $A + C$ is monotone, we have

$$0 \leq \langle x_A^k - e_A^k - x^*, u_A^k + C(x_A^k - e_A^k) - (u_A^* + Cx^*) \rangle. \tag{37}$$

It follows from that $B$ is uniformly monotone, then there exists an increasing function $\Phi_B : R_+ \to [0, +\infty)$ with $\Phi_B(0) = 0$ such that

$$\Phi_B(\|x_B^k - e_B^k - x^*\|) \leq \langle x_B^k - e_B^k - x^*, u_B^k - u_B^* \rangle. \tag{38}$$

Multiple (37) and (38) by $\gamma$, respectively, and add together, we obtain

$$\gamma \Phi_B(\|x_B^k - e_B^k - x^*\|) \leq \gamma \langle x_B^k - e_B^k - x^*, u_B^k - u_B^* \rangle$$
$$+ \gamma \langle x_A^k - e_A^k - x^*, u_A^k + C(x_A^k - e_A^k) - (u_A^* + Cx^*) \rangle. \tag{39}$$

Next, we prove that $\|x_B^k - e_B^k - x^*\| \to 0$ as $k \to +\infty$. The technical proof is similar to (a). We start from estimating the right side of (39). In fact, we have

$$\gamma \langle x_B^k - e_B^k - x^*, u_B^k - u_B^* \rangle + \gamma \langle x_A^k - e_A^k - x^*, u_A^k + C(x_A^k - e_A^k) - (u_A^* + Cx^*) \rangle$$
$$= \gamma \langle x_B^k - e_B^k - (x_A^k - e_A^k), u_B^k - u_B^* \rangle + \gamma \langle x_A^k - e_A^k - x^*, u_B^k - u_B^* \rangle$$
$$+ \gamma \langle x_A^k - e_A^k - x^*, u_A^k + C(x_A^k - e_A^k) - (u_A^* + Cx^*) \rangle$$
$$= \gamma \langle x_B^k - e_B^k - (x_A^k - e_A^k), u_B^k - u_B^* \rangle + \gamma \langle x_A^k - e_A^k - x^*, u_B^k + u_A^k + C(x_A^k - e_A^k) \rangle \tag{40}$$
$$= \langle x_B^k - x_A^k + e_A^k, \gamma u_B^k - \gamma u_B^* \rangle - \langle e_B^k, \gamma u_B^k - \gamma u_B^* \rangle$$
$$+ \langle x_A^k - e_A^k - x^*, x_B^k - x_A^k + e_A^k + e_B^k - \gamma(Cx_B^k + e_C^k) + \gamma C(x_A^k - e_A^k) \rangle$$
$$= \langle x_B^k - x_A^k + e_A^k, \gamma u_B^k - \gamma u_B^* + x_A^k - e_A^k - x^* \rangle + \langle e_B^k, -\gamma u_B^k + \gamma u_B^* + x_A^k - e_A^k - x^* \rangle$$
$$+ \langle x_A^k - e_A^k - x^*, \gamma C(x_A^k - e_A^k) - \gamma(Cx_B^k + e_C^k) \rangle.$$

For the first term of the right side (40), we have

$$\langle x_B^k - x_A^k + e_A^k, \gamma u_B^k - \gamma u_B^* + x_A^k - e_A^k - x^* \rangle$$
$$= \langle x_B^k - x_A^k + e_A^k, x_A^k - x_B^k + e_B^k - e_A^k + z^k - z^* \rangle \to 0, \tag{41}$$

as $k \to +\infty$. For the second term of the right side (40), we have

$$\langle e_B^k, -\gamma u_B^k + \gamma u_B^* + x_A^k - e_A^k - x^* \rangle$$
$$= \langle e_B^k, x_A^k + x_B^k - e_A^k - e_B^k - z^k + z^* - 2x^* \rangle \to 0, \tag{42}$$

as $k \to +\infty$. Since $C$ is $\frac{1}{\beta}$-Lipschitz continuous, we have

$$\|\gamma C(x_A^k - e_A^k) - \gamma(Cx_B^k + e_C^k)\| \leq \gamma \frac{1}{\beta} \|x_A^k - e_A^k - x_B^k\| + \gamma \|e_C^k\| \to 0, \tag{43}$$

as $k \to +\infty$. From (39)–(43), it follows that

$$\|x_B^k - e_B^k - x^*\| \to 0, \tag{44}$$

as $k \to +\infty$. Therefore, we have $\|x_B^k - x^*\| \to 0$ as $k \to +\infty$. In fact, we have

$$\|x_B^k - x^*\| \leq \|x_B^k - e_B^k - x^*\| + \|e_B^k\| \to 0, \tag{45}$$

as $k \to +\infty$. Finally, $\|x_A^k - x^*\| \to 0$ as $k \to +\infty$ is due to the fact that $\|x_A^k - x_B^k\| \to 0$ as $k \to +\infty$.

(c) By (4) and (5), we know that $Cx_B^k \to Cx^*$ and $x_B^k \rightharpoonup x^*$ as $k \to +\infty$. Since $C$ is demiregular, $x_B^k \to x^*$ as $k \to +\infty$. Also, we have $x_A^k \to x^*$ as $k \to +\infty$. This completes the proof. $\square$

Let $B \equiv 0$ and $e_B^k = 0$. Then the inexact three-operator splitting algorithm (Algorithm 1) reduces to the *inexact forward-backward splitting algorithm* as follows:

$$\begin{cases} x_A^k = J_{\gamma A}(z^k - \gamma(Cz^k + e_C^k)) + e_A^k, \\ z^{k+1} = z^k + \lambda_k(x_A^k - z^k), \end{cases} \tag{46}$$

which is equivalent to

$$z^{k+1} = (1 - \lambda_k)z^k + \lambda_k(J_{\gamma A}(z^k - \gamma(Cz^k + e_C^k)) + e_A^k). \tag{47}$$

From Theorem 1, we obtain the following convergence results of the inexact forward-backward splitting algorithm immediately.

**Corollary 1.** *Let $A : H \to 2^H$ be a maximally monotone operator. Let $C : H \to H$ be a $\beta$-cocoercive operator, for some $\beta > 0$. For any $z^0 \in H$, let $\{z^k\}$ be defined by (46). Assume that $\gamma \in (0, 2\beta)$ and $\lambda_k \in (0, \frac{4\beta-\gamma}{2\beta})$ such that $\sum_{k=0}^{+\infty} \lambda_k(\frac{4\beta-\gamma}{2\beta} - \lambda_k) = +\infty$. Let $\{e_A^k\}$ and $\{e_C^k\}$ be absolutely summable sequences in $H$. Then the following hold:*

(1)  *$\{z^k\}$ converges weakly to a solution $x \in zer(A + C)$;*
(2)  *$\|Cz^k - Cx\| \to 0$ as $k \to +\infty$ for $\lambda_k \geq \underline{\lambda} > 0$;*
(3)  *$\|J_{\gamma A}(z^k - \gamma Cz^k) - z^k\| \to 0$ as $k \to +\infty$;*
(4)  *Let $\lambda_k \geq \underline{\lambda} > 0$ and let $z^* \in zer(A + C)$. Suppose that one of the following conditions holds:*

   (a)   *$A$ is uniformly monotone on every nonempty bounded subset of dom $A$;*
   (b)   *$C$ is demiregular at each point $x \in zer(A + C)$.*

*Then the sequence $\{x_A^k\}$ converge strongly to a solution of $zer(A + C)$.*

**Remark 1.** *Corollary 1 provides a larger choice for the relaxation parameters $\{\lambda_k\}$ than Corollary 6.5 of Combettes [36] when the stepsizes remain constant in [36]. In addition, Corollary 1 also generalizes Theorem 3.4 of Combettes and Wajs [26] for solving the convex minimization probelm of the sum of two convex functions (i.e., $h(x) = 0$ in (2)) to the general monotone inclusion problem (i.e., $B = 0$ in (1)).*

Let $C \equiv 0$ and $e_C^k = 0$. Then the inexact three-operator splitting algorithm reduces to the *inexact Douglas-Rachford splitting algorithm* as follows:

$$\begin{cases} x_B^k = J_{\gamma B}(z^k) + e_B^k, \\ x_A^k = J_{\gamma A}(2x_B^k - z^k) + e_A^k, \\ z^{k+1} = z^k + \lambda_k(x_A^k - x_B^k), \end{cases} \tag{48}$$

which is equivalent to

$$z^{k+1} = z^k + \lambda_k(J_{\gamma A}(2(J_{\gamma B}(z^k) + e_B^k) - z^k) + e_A^k - (J_{\gamma B}(z^k) + e_B^k)). \tag{49}$$

Now, we have the convergence results of the inexact Douglas-Rachford splitting algorithm from Theorem 1 as follows:

Define an operator $T := I - J_{\gamma B} + J_{\gamma A}(2J_{\gamma B} - I)$, where the operator $T$ is usually called the *Douglas-Rachford splitting operator*.

**Corollary 2.** *Let $A : H \to 2^H$ and $B : H \to 2^H$ be maximally monotone operators. For any $z^0 \in H$, let $\{z^k\}$ be defined by (48). Assume that $\gamma > 0$, and $\lambda_k \in (0, 2)$ such that $\sum_{k=0}^{+\infty} \lambda_k(2 - \lambda_k) = +\infty$. Let $\{e_A^k\}$ and $\{e_B^k\}$ be absolutely summable sequences in H. Then the following hold:*

(1)　*$\{z^k\}$ converges weakly to a fixed point of T;*

(2)　*$\|J_{\gamma A}(2J_{\gamma B}(z^k) - z^k) - J_{\gamma B}(z^k)\| \to 0$ as $k \to +\infty$;*

(3)　*Let $\lambda_k \geq \underline{\lambda} > 0$ and $z^*$ be a fixed point of T. Then the iterative sequence $\{x_B^k\}$ converges weakly to $J_{\gamma B}z^* \in zer(A + B)$;*

(4)　*Let $\lambda_k \geq \underline{\lambda} > 0$ and $z^*$ be a fixed point of T. Then the iterative sequence $\{x_A^k\}$ converges weakly to $J_{\gamma B}z^* \in zer(A + B)$;*

(5)　*Let $\lambda_k \geq \underline{\lambda} > 0$ and let $z^* \in zer(A + B)$. Suppose that one of the following conditions holds:*

　　(a)　*A is uniformly monotone on every nonempty bounded subset of dom A;*

　　(b)　*B is uniformly monotone on every nonempty bounded subset of dom B.*

*Then the sequence $\{x_A^k\}$ and $\{x_B^k\}$ converge strongly to a solution of $zer(A + B)$.*

**Remark 2.**

(I)　*(1) of Corollary 2 recovers Corollary 5.2 of [36].*

(II)　*(2)–(5) of Corollary 2 generalize Theorem 25.6 of Bauschke and Combettes [53] from the exact Douglas-Rachford splitting algorithm to the inexact Douglas-Rachford splitting algorithm.*

## 4. Conclusions

In this paper, we generalized the three-operator splitting algorithm proposed by Davis and Yin [25] from exact to inexact. The theoretical convergence of the inexact three-operator splitting algorithm was studied under mild conditions on the parameters. In the forthcoming works, we will discuss the convergence rates of the inexact three-operator splitting algorithm including the fixed point residual and the ergodic and the nonergodic convergence rates of the function values in the context of convex optimization problems.

**Author Contributions:** C.Z., Y.T. and Y.J.C. contributed equally in this work.

**Funding:** This research was funded by the National Natural Science Foundations of China grant number 11661056,11401293.

**Acknowledgments:** We would like to thank the associate editor and the two reviewers for their helpful comments to improve the paper.

**Conflicts of Interest:** The authors declare no conflict of interest.

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
