# Peer review of "Convergence Analysis of an Inexact Three-Operator Splitting Algorithm"

_symmetry, doi:10.3390/sym10110563_

Reviewer 1 Report

1.    It is not clear how how the equations (3.6) were obtained.

2.    Also, it is not clear the equivalence between (3.36) and (3.37).

3.    In Remark 3.2 it is used the expression “the exact Douglas-Rachford splitting algorithmwhich is not defined. Only the notion “the inexact Douglas-Rachford splitting algorithm” is defined.

4.    Punctuations are used randomly. Insert comma or full stop after each and every equation accordingly.

5.    Must be corrected some "glitches" of editing. See, for instance, Definition 2.1.

6.    References are not uniformly written. In some references the name of the  journal  is written in full and in others it is abbreviated. In some cases, the abbreviation is not standard.

7. Also, the authors  must strengthen the References section with some articles that use some similar techniques, for instance: Weak Solutions in Elasticity of Dipolar Porous Materials, Math Probl Eng, 2008 (2008), pp. 1-8, Art. No. 158908;

Author Response

 Dear Editors and Reviewers:

Thank you for your kind letter! Thank you for the reviewers’ comments concerning our manuscript entitled ``Convergence analysis of an inexact three-operator splitting algorithm" (Manuscript ID: symmetry-379401). Those comments are all valuable and very helpful for revising and improving our paper.

We have studied comments carefully and have made the correction which we hope meet with approval.

Reply to Reviewer:

1. For the equation (3.6), the first inequality in (3.6) is come from the Cauchy-Schwartz inequality. The Second inequality in (3.6) is due to the fact that the operator $(1-\lambda_k )I+\lambda_k T$ is nonexpansive and $z^{*}$ is a fixed point of $T$, then, we have

\begin{align*}

 \| (1-\lambda_k)(z^k-z^*) + \lambda_k (Tz^k - z^*) \| & = \| (1-\lambda_k)z^k + \lambda_k Tz^k - ( (1-\lambda_k)z^* + \lambda_k T z^*)  \| \nonumber \\

 & \leq \|z^k-z^*\|.

\end{align*}

From the first part of Theorem 3.1, we know that $\{\|z^k-z^*\|\}$ is bounded, then we can get the result of the second inequality. The last equality in (3.6) comes from Lemma 2.5.

2. The equation (3.37) is come from by substituting $x_{A}^{k}$ into $z^{k+1}$  in (3.36). Then, we obtain the equivalence between (3.36) and (3.37).

3. The exact Douglas-Rachford splitting algorithm is obtained by letting the error sequences $e_{B}^{k}$ and $e_{A}^{k}$ equally to zeros. We are sorry for our negligence of explaining. In the literature, researchers just say the Douglas-Rachford splitting algorithm without the word of exact.

4. We have insert comma and the full stop after each equation, which is absent before.

5. We are sorry for our incorrect writing some expressions in Definition 2.1. We have corrected it. At the same time, we also checked all definitions and lemmas in Section 2 to ensure their correct.

6. As Reviewer pointed out, we have corrected the wrong written of the references. According to the references style, we modified the format of the corresponding references.

7. We added some references as Reviewer suggested. For example,

Weak solutions in elasticity of dipolar porous materials. Math. Probl. Eng.

2008, 2008, 158908.

 Convex optimization problem prototyping for

image reconstruction in computed tomography with the Chambolle-Pock algorithm. Phys.

Med. Biol. 2012, 57, 3065-3091.

The theory of generalized thermoelasticity with fractional order

strain for dipolar materials with double porosity. J. Mater. Sci. 2018, 53, 3470-3482.

We tried our best to improve the manuscript and made some changes based on the Reviewer's report. These changes will not influence the content and framework of the paper.

We appreciate for Editors and Reviewers’ warm work earnestly, and hope that the correction will meet with approval.

Once again, thank you very much for your comments and suggestions.

Thank you!

Yours sincerely,

Yuchao Tang

Reviewer 2 Report

The paper concerns the three-operator splitting technique. This is a new method in operator theory, that has recently received great attention. The authors propose here a generalization of the method, including the so-called inexact case. The setting is very general, in the frame of Hilbert spaces. The results are new and the proofs are correct and very precise. In particular the results of convergence are potentially useful for applications. The bibliography is exhaustive, including a list of the papers devoted to the subject in these last years. Summing up, the paper should be published in the present form.

Author Response

Dear Editors and Reviewers:

Thank you for your kind letter! Thank you for the reviewers’ comments concerning our manuscript entitled ``Convergence analysis of an inexact three-operator splitting algorithm" (Manuscript ID: symmetry-379401). Those comments are all valuable and very helpful for revising and improving our paper.

We have studied comments carefully and have made the correction which we hope meet with approval.

We would express us thank you to the reviewer's comments concerning our paper.

We tried our best to improve the manuscript and made some changes based on the Reviewer's report. These changes will not influence the content and framework of the paper.

We appreciate for Editors and Reviewers’ warm work earnestly and hope that the correction will meet with approval.

Once again, thank you very much for your comments and suggestions.

Thank you!

Yours sincerely,

Yuchao Tang

Round  2

Reviewer 1 Report

The authors considered all my suggestions, which led to an improved form of the manuscript.